# Coffee Rust Forecast Systems: Development of a Warning Platform in a Minas Gerais State, Brazil

**Edson Ampélio Pozza** [1,*] **, Éder Ribeiro dos Santos** [2]**, Nilva Alice Gaspar** [2]**, Ximena Maira de Souza Vilela** [3]**, Marcelo de Carvalho Alves** [4] **and Mário Roberto Nogueira Colares** [1]

1 Laboratory of Epidemiology and Plant Disease Management, Department of Phytopathology, Federal University of Lavras, Lavras 37200-900, MG, Brazil; mnogueiracolares@gmail.com
2 Department of Geoprocessing, Cooperative of Coffee Growers in Guaxupé-COOXUPÉ, Guaxupé 37800-000, MG, Brazil; edersantos@cooxupe.com.br (É.R.d.S.); nilva_alice@hotmail.com (N.A.G.)
3 Manager of Fungicides, Iharabras Chemical Industries-IHARA, Sorocaba 18087-170, SP, Brazil; ximena.vilela@ihara.com.br
4 Department of Engineering, Federal University of Lavras, Lavras 37200-900, MG, Brazil; marcelo.alves@ufla.br
* Correspondence: edsonpozza@gmail.com

**Abstract:** This study aimed to develop a warning system platform for coffee rust incidence fifteen days in advance, as well as validating and regionalizing multiple linear regression models based on meteorological variables. The models developed by Pinto were validated in five counties. Experiments were set up in a randomized block design with five treatments and five replications. The experimental plot had six lines with 20 central plants of useful area. Assessments of coffee rust incidence were carried out fortnightly. The data collected from automatic stations were adjusted in new multiple linear regression models (MLRM) for five counties. Meteorological variables were lagged concerning disease assessment dates. After the adjustments, two models were selected and calculated for five counties, later there was an expansion to include ten more counties and 35 properties to validate these models. The result showed that the adjusted models of 15–30 days before rust incidence for Carmo do Rio Claro and Nova Resende counties were promising. These models were the best at forecasting disease 15 days in advance. With these models and the geoinformation systems, the warning platform and interface will be improved in the coffee grower region of the south and savannas of the Minas Gerais State, Brazil.

**Keywords:** incidence; multiple linear regression models; meteorological variables; Brazil

## 1. Introduction

Since coffee is usually served hot and possesses functional compounds, such as caffeine, amino acids, sugars and phenols, it is one of the most consumed beverages worldwide [1,2]. It is consumed in all continents, but mainly in the populous northern hemisphere, which has low temperatures during most of the year. However, it is produced in the southern hemisphere, where a tropical climate predominates, without an intense cold period to contribute to the reduction of the initial inoculum. Moreover, the soils are most often poor and the raining season is mainly restricted to the summer, adding to the current extreme weather scenario. Under these conditions, producers cultivate varieties resistant to these climate conditions, that is, to drought and frost, during the winter or specific periods, and also varieties adaptable to new cultivation technologies, resulting in increased productivity [3]. Nevertheless, the high productivity associated with the favorable environment and the disease susceptibility of cultivars can lead to significant losses in the main Brazilian producing regions, principally in the Minas Gerais state [4–12]. In Brazil, loss of productivity associated with the disease can reach up to 30% [13]. However, in a recent study, Colares [14], using mathematical modeling, estimated that these yield

losses might vary from 59.8% to 99.8% over more than one year of cultivation. Rust causes deleterious effects on the coffee tree [15], including defoliation, which is responsible for reducing the tree photosynthetic area, and consequently death of the plagiotropic branches with reflexes in posterior crop loads [6,16,17].

According to Pinto et al. [18], the first disease signs, based on the disease progress curve, occur from December to January. Then, there is an increase in the disease progress rate in March and April, assuming exponential growth. Finally, from June to August, normally after the harvest in a cold climate with reduced rainfall, the culture displays the most intense disease signs throughout the year. The pathogen sporulation reaches its greatest intensity and a drastic fall of leaves can occur with severe reduction of the plant canopy.

High crop load, susceptible cultivars, dense plantation, temperature between 21 and 25 °C with periods of leaf moisture between 6 and 24 h [15,19–22], shading, nutritional imbalance and water deficit increase the high disease progress rate (r) [6]. The nutritional balance of the coffee crop makes it more difficult to achieve the highest productivity, as a result of the drain of plant nutrients and soil reserves, causing, therefore, a tendency to increase the disease intensity [6]. Under these growing conditions, to mitigate the risks of epidemics with high rates of rust progress (r), the maintenance of water and soil fertility, which influence the nutrition of the coffee trees and consequently constitute and reinforce resistance barriers against the pathogen, should be part of the disease management strategies [23–30]. Furthermore, periodic pruning and the use of disease resistant or tolerant varieties contribute to the disease control [31].

In addition to management techniques, fungicides are also used [6,32]. Currently, scheduled sprayings are carried out with protective and/or systemic fungicides in the months with the greatest disease progress rate (r) from the very beginning of the rainy summer in Brazil. In any case, they must be applied when the visual sign or manifestation of the pathogen sporulation in the leaves is still below a 5% incidence. These sporulations indicate that other points of infection and/or colonization that has not yet been spotted already exist, which characterizes coffee rust. In order to avoid the pressure to select resistant populations of *H. vastatrix*, rust control programs are used, where two or three sprays of a mixture of triazoles or carboxamides with strobilurins, the strobymix, associated or alternated with two or three applications of protectors, cuprics or dithiocarbamates, during the rainy season [6].

However, consumers worldwide are demanding better environmental, social and economic sustainability in coffee plantations by reducing and optimizing fungicide spraying on crops [29].

In terms of disease management, there is an ongoing search by coffee growers for a more effective method of disease control that can reduce crop maintenance costs and increase profit, generating financial resources to be invested in the environmental and social sustainability of the coffee farms and, consequently, attaching the producers to their rural properties. This behavior conforms with consumer demands on a global scale. Therefore, to avoid a fixed spraying schedule and applications on dates unfavorable to rust or after infection and colonization of the pathogen, warning or forecast systems could be used [33]. According to Pinto et al. [18] and Hinnah et al. [34], using variables from the disease triangle, that is, the pathogen, the host, or the environment, the disease forecast systems can be used as a tool to contribute and guide crop disease management and the rational use of fungicides, and, in this way, reaching global demands to mitigate risks to the environment, in the context of green thinking. Therefore, environmental sustainability could be balanced with the economic and social needs of the agricultural production [35–37].

The main objectives of the disease forecast system are to optimize or increase efficiency of fungicide applications, decreasing production cost and the risk of economic losses caused by diseases and consequently increasing productivity [33,34,38,39]. The cost–benefit ratio is another criterion mentioned by Campbell and Madden [33] for early warning or forecast systems. According to the authors, these procedures should be available at reasonable

costs, especially considering the potential savings in management costs and reduction of losses caused by the disease. Thus, the platform or software must be simple and proven efficient, with an interface for technicians and producers easy to implement in the field, and understandable for the coffee grower to improve the spray efficiency.

Warning or forecast systems are decision support methods or tools to help farmers define the best moment to apply the techniques to control plant diseases [40]. Thus, there is a spray optimization with greater efficiency in pest control and increased productivity since the risks of epidemics are continuously monitored. Therefore, fungicide sprayings for disease management are carried out only under environmental conditions favorable to their progress [33], when hosts or even populations of pathogenic and virulent agents are susceptible. Thus, unnecessary sprays, earlier or later than the infection period are not carried out, optimizing the use of sprayings and the producer's time dedicated to this activity [41]. Furthermore, there is a reduction in the probability of contamination of workers responsible for directly handling the products and a reduction in the environmental impacts as well [42].

These systems use information from the environment and/or host and/or pathogen and/or disease to warn the producers of future disease intensity values [33]. Many early warning systems use information from the environment to estimate future disease intensity values because the environment can determine whether epidemics occur [18,43–48]. The occurrence of the disease process depends on the minimum duration of favorable environmental conditions to the establishment of parasitic relationships between the pathogen and its host, known as the critical period. If this is not met, phases of the pathogen's life cycle such as germination, infection and colonization may be compromised. The most important variables are air temperature and water availability at infection sites, which are characterized by the relative humidity of the air and the duration of the leaf moisture period, caused by rain, dew, fog or irrigation. Such variables are strongly associated with the progress of plant diseases [49] and, therefore, are often used in estimating future disease values [41].

Traditionally, forecast or warning systems for coffee diseases have been based on the use of regression models or double-entry variable tables, mainly using temperature and water availability as variables [15,18,38,50]. However, with the current extreme weather scenario, the cultivation of coffee in different biomes and edaphoclimatic conditions and the increase in productivity and new techniques of cultivation and management of the coffee tree, other approaches, such as time series techniques, neural networks, Bayesian network, fuzzy logic, decision trees, and others, have been gaining the attention of researchers [18,51]. These new tools, together with traditional techniques, can help one understand the dynamics of the disease process, providing better results in disease control.

The variables used to build up forecasting systems in this new scenario can be selected from mathematical and statistical tools, besides the use of artificial intelligence, according to the characteristics of the pathosystem to which they are related. The variables selected by these methods, which belong to the disease triangle, that is, pathogen, environment and host [33], were used in several studies to build and validate disease prediction systems for coffee crops [34,52–54].

The reason for using forecasting systems involves economic, environmental and disease-related aspects [33]. Thus, coffee rust, nowadays, in a scenario of climate change and the protection of rural workers and the environment, meets important prerequisites for developing a forecasting system. Furthermore, it is an economically important disease that does not occur on established dates to provide a calendar for spraying fungicides, but occurs seasonally or sporadically, a recurrent factor in the current scenario of extreme weather, and it is cultivated in different biomes and under high pressure to increase productivity, besides having traditional control that still results in a cost to the producer.

Although warning systems were developed to monitor coffee leaf rust [18,43,44,50,51,55,56] due to the importance and conditions already mentioned, few were implemented and validated for different Brazilian regions or biomes where coffee is produced.

Nowadays, in the current context of green thinking, seeking to mitigate risks to the environment, the use of forecasting or warning systems is gaining new momentum. However, the benefits for the farmers should be real, and could not be achieved without the system. Certainly, the main variables selected to calculate forecast equations have already been identified by researchers. However, before becoming available to farmers, these methodologies have to be validated in different coffee-producing regions or biomes. Few of the forecasting systems mentioned above have been validated for several Brazilian regions or biomes different from those where they have been evaluated. Concomitantly, the hardware and methodology must supply the software daily, and the consultation or warning platform must be set up on an easy-to-interpret interface for producers and technicians.

Besides being the largest coffee producer, Brazil is a country with continental dimensions with different edaphoclimatic conditions in coffee-producing regions, presenting different altitudes, landscapes, climates and soil types, among other distinct variables. The continental dimensions and edaphoclimatic variability of the country are a hindrance to validating forecasting systems in regions different from where they were initially developed and evaluated.

Validation in different regions is the differential of the forecast system to be proposed in relation to the others mentioned above. In addition, it will be regionalized, with selected climate and host variables, in the main coffee regions of the Minas Gerais state, Brazil. Thus, it is expected to have its efficiency and applicability in the field proven, helping the producer to control coffee rust in a sustainable way and meeting the green thinking of globalized consumers. In this way, it is not only about producing science, but offering technology, with applicability with the producer.

In light of the above, based on the state of the art presented, this study aims to develop a warning or forecast platform for the coffee rust incidence, 15 days in advance, to validate and regionalize Multiple Linear Regression Models from regionalized meteorological variables for the main coffee producing regions and biomes in the state of Minas Gerais, Brazil.

## 2. Methodology to Construct Forecast Models and Interface

The use of predictive models is an alternative to optimize and rationalize the use of fungicides. Therefore, we evaluated the viability of two prediction models developed by Pinto et al. [18]. Since the forecast models were developed with meteorological data from Lavras county, Minas Gerais state, Brazil, other geographic locations were incorporated to validate the models. Then, the study was divided into three phases. In phase I, five sampling points were initially implemented in different counties from 2018 to 2019, using two warning or forecasting methodologies, collecting meteorological variables and assessing coffee rust. In phase II, in 2020, ten more counties were incorporated into the system, with the selection of variables and the adjustment of the multiple regression model from the data collected in the five previous counties. In phase III, besides those fifteen locations, another 35 properties for the collection of meteorological and disease data were incorporated, totaling 50 sampling points to obtain disease and host data in the state of Minas Gerais, Brazil.

### 2.1. Phase 1: Validation of the Forecast System from 2018 to 2020

Initially, a total of five areas (Table 1) with trials were used to validate the forecasting systems from models developed [18], which still have the system operating and were chosen for being located at different altitudes and producing regions in Alto Paranaíba, in the south and the north of the state.

**Table 1.** Location of areas for validation of forecast systems in five municipalities in the state of Minas Gerais, Brazil.

| Municipality | Location | Altitude (m) | Latitude (S) | Longitude (W) |
|---|---|---|---|---|
| Carmo do Rio Claro | Fazenda Boa Esperança | 796 | 21.004600 | 46.022900 |
| Monte Santo de Minas | Sítio Bela Vista | 915 | 21.181100 | 46.965600 |
| Nova Resende | Sítio São João | 1184 | 21.104300 | 46.410400 |
| Rio Paranaíba | Fazenda Caetés e Olhos D'água | 1129 | 19.226100 | 46.219200 |
| Serra do Salitre | Fazenda Cachoeira do Campo | 1200 | 19.163300 | 46.589200 |

In this phase, these areas were conducted with two consecutive harvests, 2018/2019 and 2019/2020, to validate the forecast models and obtain data on meteorological and host variables. Rust was evaluated fortnightly during this period, seeking to add the meteorological variability [6] and the biennial of coffee production [57,58]. The arabica coffee (*Coffea arabica* L.) planted in these fields was an around-seven-year-old variety of the cultivar Catuaí group with red fruits, rust susceptible, in a spacing of 3.8 m × 0.6 m between rows and plants, respectively. The tests were conducted according to the technical recommendations for coffee growing in Brazil [59–64]. Weed and pest management was carried out according to the control level. The soil fertility management and crop nutrition was performed based on the results of chemical analysis of the soil and nutrition of leaves, applying correctives and fertilizers in the projection of the plant canopy, according to Mesquita et al. [61], Alvarez V.; Ribeiro [65] and Alvarez V. et al. [66].

Standard disease and pest controls were carried out in all experimental areas for uniformization before starting the trial programs to validate the warning systems. For disease control, the fungicide Epoxiconazole (50 g L$^{-1}$ of the active ingredient) + Fluxapyroxade (50 g L$^{-1}$ of the active ingredient) + Pyraclostrobin (81 g L$^{-1}$ of the active ingredient), trade name Ativum$^{®}$, was used at a dose of 1.5 L ha$^{-1}$ in October 2018.

### 2.1.1. Sampling of Environmental Variables

The regression equations were supplied with environmental data obtained from the automatic stations model Davis Vantage Pro$^{®}$ in the counties and transferred to the Weather Link$^{®}$ software located in the Cooperative of Coffee Growers in Guaxupé (Cooxupé) (Table 1). The data were recorded on an hourly basis and sent via a dedicated link to the server system at Cooxupé's headquarters in Guaxupé, Minas Gerais, where they were processed and entered into an interface designed to visualize the results.

### 2.1.2. Forecasting Systems and Spraying Methodology

Two phytosanitary warning systems were evaluated during two consecutive harvests (2018/2019 and 2019/2020). Initially, the system developed by Pinto et al. [18] was validated when the fortnightly assessment of coffee rust incidence reached 5% (a control level also called the technical assessment model). In the formula used by Pinto et al. [18], the warnings were issued when the model calculated a rust incidence of 5% or more. The models were called Decision Support System 1 (DSS 1–5% control level also called the technical assessment model) and 2 (DSS 2) (Equation (1)), respectively:

$$y = -39.12 - 1.09 * Tavg_{30} + 0.30 * DP_{45} + 0.54 * RHavg_{60} + 2.68 * NIH_{60} \qquad (1)$$

where:

$y$ = Rust incidence forecast;
$Tavg_{30}$ = Average of mean temperatures to 30 days before rust incidence;
$DP_{45}$ = Days with precipitation to 45 days before rust incidence: precipitation > 0 mm;
$RHavg_{60}$ = Average of mean relative humidity to 60 days before rust incidence;
$NIH_{60}$ = Average of mean number of isolation hours to 60 days before rust incidence.

The experiments were implemented in a randomized block design (RBD) with five treatments (Table 2) and five replications. The experimental plot had six lines with 20 plants

each, totaling 120 plants. Two lines were discarded on each side and in the two inner parts were 10 plants, five at each end. Therefore, the useful area plot for evaluation contained 20 inner plants.

**Table 2.** Treatments, trade name, dose per hectare, active ingredient and their concentration and chemical group, used in the areas for validation of the warning systems in the 2018/2019 and 2019/2020 harvests.

| Treatments | Trade Name | Dose per Hectare | Active Ingredient | Concentration of Active Ingredient | Chemical Group |
|---|---|---|---|---|---|
| 1. Control | - | - | - | - | - |
| 2. Standard farm | Verdadero® and Opera® (with 2 pulverization) or Priori Xtra, depends of the location | 1 kg + 1.5 L + 0.75 L | Cyproconazole + ThiamethoxamPyraclostrobin + Epoxiconazole and Azoxystrobin + Cyproconazole | $300 \text{ g kg}^{-1}$ + $300 \text{ g kg}^{-1} \text{e} 133 \text{ g L}^{-1}$+ $50 \text{ g L}^{-1}$ and $200 \text{ g L}^{-1}$+ $80 \text{ g L}^{-1}$ | Triazole + Neonicotinoid and Strobirulin |
| 3. IHARA | Fusão® and Spirit® | 1.5 L + 2 L | Tebuconazole + Metominostrobin and Flutriafol + Dinotefuran | $165 \text{ g L}^{-1}$ + $110 \text{ g L}^{-1}$ and $273 \text{ g L}^{-1}$ + $87.5 \text{ g L}^{-1}$ | Triazole + Strobirulin + Neonicotinoid |
| 4. DSS 1 [1] | Fusão® | 1.5 L | Tebuconazole+ Metominostrobin | $165 \text{ g L}^{-1}$ + $110 \text{ g L}^{-1}$ | Triazole + Strobirulin |
| 5. DSS 2 [1] | Fusão® | 1.5 L | Tebuconazole+ Metominostrobin | $165 \text{ g L}^{-1}$ + $110 \text{ g L}^{-1}$ | Triazole + Strobirulin |

[1] DSS—Decision support system.

As of November 1, 2018, to control rust, when the warnings for DSS 1 and 2 were issued, a systemic fungicide mixed with a mesostemic one was used (Table 2). Spraying took place using a manual costal sprayer model Jacto PJB 20® in a volume of 400 L of water $\text{ha}^{-1}$.

The coffee rust incidence assessment was carried out fortnightly from 31 October 2018, randomly, by non-destructive method, on five plagiotropic branches per plant, on the morning shade side, in the middle third of the plants [67]. In each of the 20 plants, five leaves were evaluated from the second to fourth pairs of nodes in the plagiotropic branch, totaling 100 leaves per plot or replicate. New sporulating lesions with a light-yellow appearance characteristic of the pathogen signs were observed.

For this, the rust incidence was determined by the percentage of the number of leaves with damage in relation to the number of leaves evaluated, through the following equation [33]:

$$I(\%) = \left( \frac{NLL}{NTL} \right) * 100 \qquad (2)$$

where:

$I(\%)$ = Coffee rust incidence;
$NLL$ = Number of lesioned leaves;
$NTL$ = Number of total leaves sampled on the coffee tree.

2.1.3. Development of the Interface to View Phytosanitary Warnings or Coffee Rust Forecast

After the collection of meteorological variables from the meteorological stations, they were transmitted, received and processed. The resulting spraying warning was made available in a proper interface for the five counties located in the Southeast region of Brazil in Minas Gerais state. For the interface development, the following programming languages were used: Python 2.7 with the Django Framework 1.6 (back-end) and HTML, CSS, JavaScript, Jquery with the Bootstrap Framework (front-end). The latter is responsible for the entire presentation of the system, as well as its responsiveness (adaptability to

different screen sizes: TV, notebook, tablet, smartphone). In parallel to this system, routines were also developed using the Java SE programming language.

Thus, using two warning methodologies, DSS 1 and 2, the spraying warning was issued when the predicted or estimated control level reached an incidence of 5%. The technicians and their supervisors received the spraying warning through email. After spraying, they informed the system in the built interface, and then the daily emails stopped being sent. After three days, besides sending spraying warnings, another daily email was sent to notify the technicians and supervisor in charge of the area about the spray delay.

*2.2. Phase 2: Adjustment of Models with Data Collected in Five Different Counties in the State of Minas Gerais, Brazil*

In this phase, new regression equations were developed with the disease incidence as a function of the variables collected in the meteorological stations of the five counties mentioned above. Thus, 18 variables were generated (Table 3).

**Table 3.** Environmental variables collected and generated with an automatic station model Davis Vantage Pro®, software Weather Link®, located in the five cities or nuclei of the Cooperative of Coffee Growers of Guaxupé (Cooxupé).

| Variables | Description |
|---|---|
| Tavg [1] | Average of mean temperatures |
| Tmax [1] | Average of maximum temperatures |
| Tmin [1] | Average of minimum temperatures |
| RHavg [1] | Average of mean relative humidity |
| RHmin [1] | Average of minimum relative humidity |
| RHmax [1] | Average of maximum relative humidity |
| WS [1] | Windy speed |
| IH [1] | Insolation hours |
| DPT [1] | Dew point temperature |
| LT [1] | Leaf temperature |
| WH [2] | Wetness hours |
| P [2] | Precipitation |
| $TavgLW_{(6 \text{ p.m.}–9 \text{ a.m.})}$ [1] | Average temperatures with leaf wetness from 6 p.m. to 9 a.m. |
| $TavgLW_{(6 \text{ p.m.}–6 \text{ a.m.})}$ [1] | Average temperatures with leaf wetness from 6 p.m. to 6 a.m. |
| $NHDT_{(\geq 18 \,°C, \, <26 \,°C)}$ [2] | Number of hours of the day with temperature $\geq 18\,°C$ and $<26\,°C$ |
| $NHDT_{(\geq 15 \,°C, \, <26 \,°C)}$ [2] | Number of hours of the day with temperature $\geq 15\,°C$ and $<26\,°C$ |
| NDP [2] | number of days with precipitation |
| $NHP_{(6 \text{ p.m.}–9 \text{ a.m.})}$ [2] | number of hours of precipitation from 6 p.m. to 9 a.m. |

[1] Variable created from the average of the values considered in the lag. [2] Variable created from the accumulated values considered in the lag.

With these data collected from October 2018 to January 2020, totaling 35 fortnightly assessments, multiple linear regression models (MLRM) of disease incidence were adjusted as a function of meteorological variables (Table 3) in each of the five counties.

These environmental variables were lagged in relation to the disease assessment dates, as follows:

1.  Average/accumulated values of 2–4, 4–7, 7–15 and 15–30 Days Before Rust Incidence (DBRI) including the day of assessment of twelve meteorological variables collected from meteorological stations from October 2018 to August 2019;
2.  Average/accumulated values of eleven meteorological variables collected from October 2018 to January 2020 at the meteorological stations fifteen days before the disease assessments, including the assessment day, according to the methodology of Pinto et al. [18] and Oliveira [50]. The adjustment of the models was performed with two data sets in and out of the harvest period (June, July and August). Regression equations were also adjusted, excluding the environmental variables Insolation Hours,

Wind Speed, Wetting and Dew Point Temperature, to obtain fitted models with few variables, also in and out of the harvest period;

3.　Average/accumulated values delayed from 15 to 45 DBRI, including the assessment day from October 2018 to January 2020, of ten meteorological variables, four of which were collected from the meteorological stations and six elaborated from these data. Initially, the best variables were selected to adjust the models. In this case, Pearson's correlation was performed between the variables and disease incidence. The analyses used significant variables and others with a correlation greater than 0.6. Afterward, the following variables were also calculated: Average of maximum, mean and minimum temperatures; Average of temperatures with leaf moisture from 6 p.m. to 9 a.m.; Average of temperatures with leaf moisture from 6 p.m. to 6 a.m.; Number of hours a day with temperature $\geq 18\,°C$ and $<26\,°C$, and $\geq 15\,°C$ and $<26\,°C$; Precipitation; Number of days with precipitation; Number of hours of precipitation from 6 p.m. to 9 a.m.. In addition, models with all these variables were fitted data in and out of the harvest period from June to August. In this case, the following variables were also excluded from the analysis: insolation hours, dew point temperature, wind speed and duration of the moisture period, which are variables obtained only from complete meteorological stations, rarely found throughout the coffee-producing areas in the state of Minas Gerais, Brazil.

After obtaining the lag of the meteorological variables, in the four periods above (item 1), they were evaluated in the MLRM, with the following general equation:

$$y = \beta_0 + \beta_1 x_1 + \beta_2 x_2 + \ldots + \beta_\rho x_\rho + \varepsilon \tag{3}$$

where:

$y$ = Disease incidence, in percentage;
$x_1, x_2, x_p$ = Environmental variables;
$\beta_0$ = Regression constant;
$\beta_1, \beta_2, \ldots, \beta_p$ = Partial regression parameters or coefficients;
$\varepsilon$ = Independent random errors.

The data were analyzed to verify if they met the assumptions of the analysis of variance, observing the Shapiro–Wilk (normality), Breush–Pagan (homoscedasticity) and Box–Pierce (independence) ($p > 0.05$) tests. Afterward, the regression analyses were performed to adjust the disease forecasting models.

To select the environmental variables, the Stepwise technique was used to estimate coffee rust incidence by MLRM. As a standard, in the estimation of the parameters of the MLRM, a least squares method was applied to minimize the sum of squares residuals [68].

The best models were selected based on the significance of meteorological variables in the $t$-test of the parameters of the regression equation ($p < 0.05$), with higher values of $R^2$ and $R^2$adjusted, and lower values of the Akaike information criterion (AIC), errors, mean of errors, standard deviations and mean squared deviations.

To assess the quality of the adjustments and identify the model with the best standard description of disease prediction, the Akaike information criterion (AIC) was applied. The AIC was based on information minimization or the Kullback–Leibler distance, being a measure of proximity between the ideal (perfect) model and the candidate. An estimate of this distance is calculated with the following equation:

$$AIC = -2\ln(L(\theta)) + 2\rho \tag{4}$$

where:

$L(\theta)$ = Estimate of the maximum likelihood function;
$p$ = Number of parameters of the evaluated model.

The adjusted model with the lowest AIC value is considered close to ideal and, therefore, the best adjustment [69].

The *lm* function was used to adjust the MLRM, and the ggplot2 package was used to make graphs and the maps were implemented in the open access software "R" version 3.6.2 [70]. After the adjusted models, the best models were validated from February to October 2020 in five counties.

### 2.3. Phase 3: Expansion of the Warning System with New Models

From October 2020, the expansion of the forecast system to ten more counties began. Of these, eight are in Minas Gerais and two in São Paulo state (Table 4), besides the previous five, totaling 15 locations, or Cooxupé branches or cooperation nucleus, with complete meteorological stations as described above. Beyond the two models selected in phase 2 to substitute DSS 1, the DSS 2 denominated Meteorological Model was also used to forecast or send the disease warning 15 days in advance in 15 counties. In this case, three equations were used to send the spray warning. The trials in the five counties of phase 1 continued to be carried out.

**Table 4.** Distribution of areas with meteorological stations for the validation of models of the coffee rust prediction system in 10 municipalities in the states of Minas Gerais and São Paulo, Brazil.

| Municipality | Altitude (m) | Latitude (S) | Longitude (W) |
|---|---|---|---|
| Alfenas [1] | 827 | 21.41373 | 45.97055 |
| Alpinópolis [1] | 935 | 20.84751 | 46.37944 |
| Cabo Verde [1] | 940 | 21.45448 | 46.41182 |
| Caconde [2] | 830 | 21.53722 | 46.63796 |
| Campestre [1] | 1082 | 21.69692 | 46.25357 |
| Campos Gerais [1] | 900 | 21.24417 | 45.75556 |
| Coromandel [1] | 962 | 18.47386 | 47.21385 |
| Guaxupé [1] | 870 | 21.28687 | 46.69303 |
| Monte Carmelo [1] | 912 | 18.75402 | 47.51819 |
| São José do Rio Pardo [2] | 755 | 21.63306 | 46.89889 |

[1] State of Minas Gerais. [2] State of São Paulo.

The fungicide spray warning in the DSS 1 treatment was issued when two of the three formulas calculated a disease incidence of 5% or more in three consecutive days. These three formulas are the two formulas selected in phase 2 (Carmo do Rio Claro 15–30 DBRI and Nova Resende 15–30 DBRI models); a third one by Hinnah et al. [34] was also incorporated. The notification methodology for DSS 2 remained the same as for phase 2.

Despite the larger error or residual of other models compared to the two selected ones, they were still used to calculate the disease incidence for further analysis, attempting to prospect or select good models, but never to send the spray warnings.

At the same time, the expansion of areas with meteorological stations, the incidence assessment in 35 other properties, was also included for the validation of these forecasting models in different regions and localities, besides the 15 already under evaluation, totaling 50 areas for rust samplings (Figure 1).

These areas are in counties situated in the savanna biome and in southern Minas Gerais state, Brazil, with atlantics forest, savanna and transitions area. In each county, four stands of coffee trees were and are still being assessed on different farms, strategically distributed and close to meteorological stations, to verify the efficiency of the proposed models in different conditions of altitude, terrain, soil, climate and crop management.

In these areas, the disease incidence was assessed in a single plot or homogeneous area of the farm on a monthly basis in 25 randomly-chosen-around-seven-year-old plants of the cultivar Catuaí group. For this sample, four leaves from the second to the fourth node of plagiotropic branches in the middle to lower third of the plant were evaluated on the morning shade side of the plantation, totaling 100 leaves per plot.

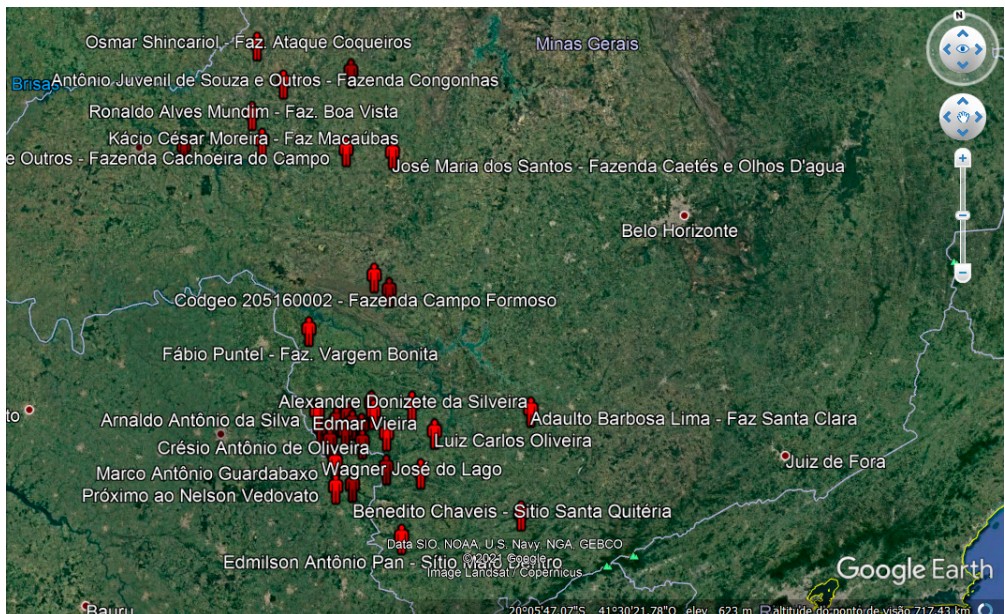

**Figure 1.** Expansion to 35 properties to validate the models developed to compose the coffee rust forecast system in the southern and savanna municipalities of the Minas Gerais state, Brazil.

Of these 50 points, the disease progress curves evaluated in the field and the estimated or predicted values from the multiple adjusted regression equations, as a function of the months of the year, were plotted to assess the quality of the adjustment as well as the forecasting system.

## 3. Results

### 3.1. Phase 1: Validation of the Models

According to the two methodologies, DSS 1 and 2, using the formulas of Pinto et al. [18] with a control level of 5% [43], the coffee rust incidence was calculated. Spray warnings were made available on the website. The five counties are located on the map of Minas Gerais (Figure 2).

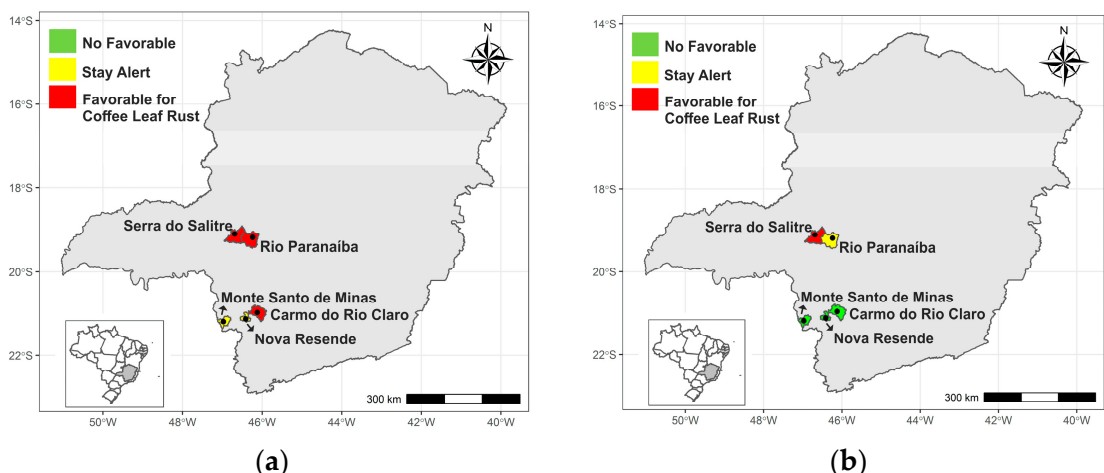

**Figure 2.** Interface for viewing the phytosanitary warning in the municipalities of Carmo do Rio Claro, Monte Santo de Minas, Nova Resende, Rio Paranaíba and Serra do Salitre in the state of Minas Gerais, Brazil: (**a**) DSS 1; (**b**) DSS 2.

The visualization and understanding of the warnings were simplified by the use of the traffic light color system, where green meant "with no probability of occurrence of rust", yellow meant "attention required with an estimated incidence of 3 to 5%, prepare the

spraying logistics, and red meant "high probability of coffee rust with a disease incidence of 5% or more after 15 days. Then spraying could be necessary".

The estimated or calculated values were compared with the real ones sampled in the field in the five locations throughout the trial period. The differences between the forecasts or calculated data and the field or real values varied among counties and months of the year (Figure 3).

The DSS 1 was replaced because in the field validation period, the 5% control level methodology presented failures in the control of the disease, when compared to the other models being validated. Furthermore, this methodology needs human resources available for the assessments every fifteen days.

### 3.2. Phase 2: Adjustment of Forecasting Models in Five Different Counties in the State of Minas Gerais, Brazil

Disease incidence models were adjusted as a function of the variables collected in the five counties, using the forecasting models proposed by Pinto et al. [18], which are based on a 5% level of disease incidence to begin the chemical control [43].

For this purpose, four models were adjusted for each county, and the best-fitted equation was selected based on the selection criteria. Models with the lagged variables 15–30 days before disease assessments were selected since they obtained higher $R^2$ and $R^2$adjusted, and lower Akaike Information Criterion (AIC) values. Furthermore, this methodology was also employed by Pinto et al. [18] since it is within the coffee rust incubation period [71–73]. However, the adjustment of these models included variables that could not be obtained from less sophisticated meteorological stations.

Therefore, after singling out the variables to obtain simpler formulas capable of being used in smaller stations of lower acquisition and maintenance value, 35 models were adjusted for the period from October 2018 to January 2020 for the five counties. From those, the ten best-fitted models were selected as described above, and then again, the two most promising models to forecast coffee rust incidence with the smallest errors or deviations were singled out, one from Carmo do Rio Claro and the other from Nova Resende, municipalities in Minas Gerais state, Brazil.

Due to better results, the DSS 1 model, from June 2020, was replaced by these other two new equations (Table 5). By replacing the meteorological variables (Table 3) used to adjust the models, it is possible to obtain forecasting values for coffee leaf rust incidence.

**Table 5.** Estimated parameters of the adjusted models for the Carmo do Rio Claro and Nova Resende municipalities, Minas Gerais state, Brazil.

| Municipality | Model [1] | Equation [2] |
|---|---|---|
| Carmo do Rio Claro | 15–30 DBRI | $Y = -304.78667634{\text{ ***}} + 13.16506156\ T_{max}{\text{ ***}} - 13.91295446\ T_{min}{\text{ ***}} - 0.07066871\ p + 3.42680986\ RH_{min}{\text{ ***}}$ |
| Nova Resende | 15–30 DBRI | $Y = 18.3098934{\text{ ***}} - 1.3387465\ T_{avg}{\text{ **}} + 1.2618530\ T_{min}{\text{ **}} - 1.0714839\ IH{\text{ **}} - 0.2205387\ WH{\text{ **}}$ |

[1] Model 15–30 DBRI: 15–30 days before rust incidence. [2] Regarding the significance of the equation parameters: ** $p < 0.01$; *** $p < 0.001$.

In an attempt to obtain better-fitted models, one of the hypotheses was to exclude the harvest period and some meteorological variables not obtained in less sophisticated meteorological stations. Thus, besides the 35 models described above, we adjusted 30 others in the same period for those five counties, and the nine best-fitted models were singled out according to the methodology described in Section 2.2. However, when implemented in the platform and interface to calculate and issue the disease control warning, those models were not promising since they displayed large errors or deviations in comparison to previous models.

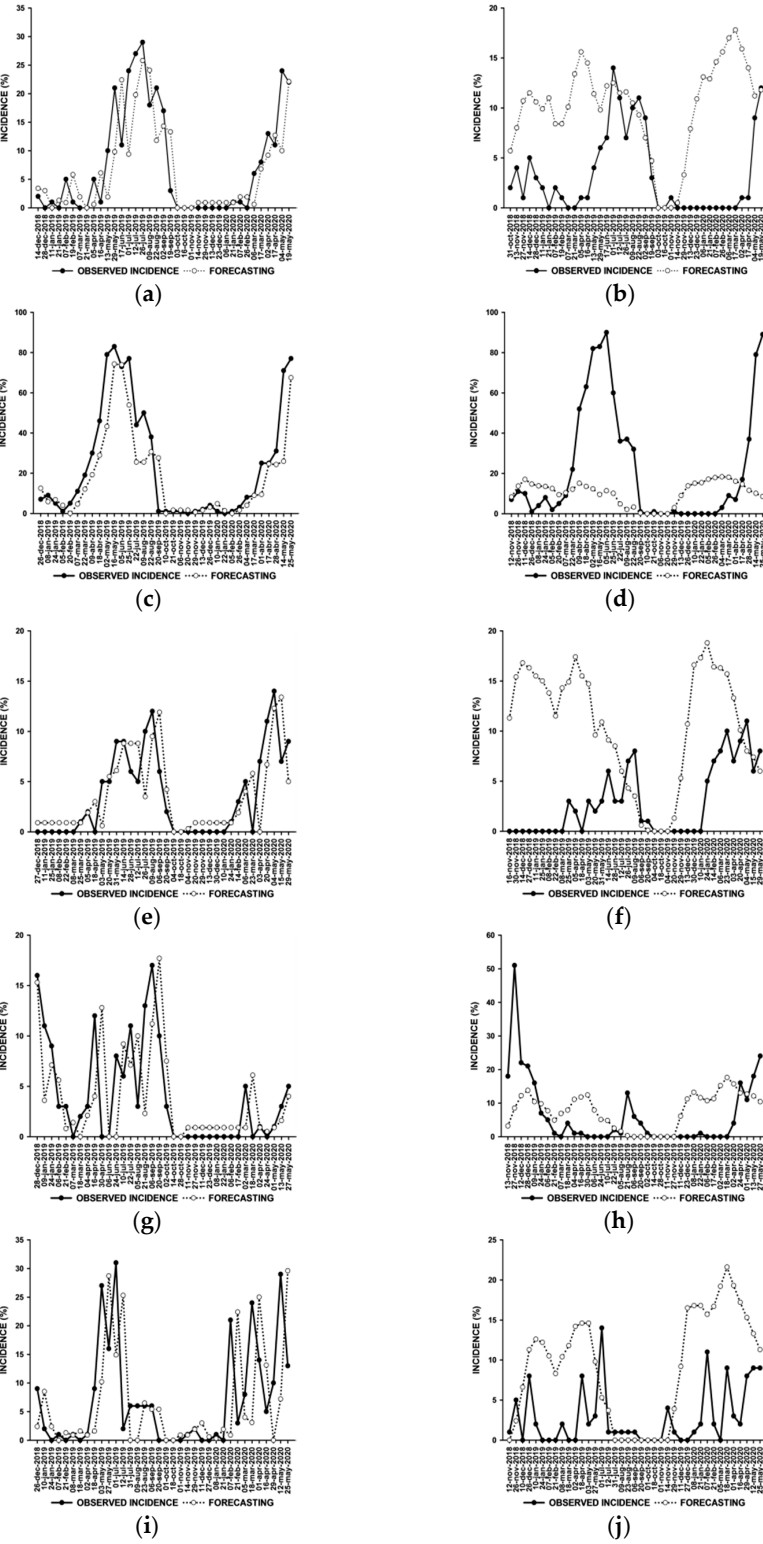

**Figure 3.** Comparison of the 5% control level methodology (DSS 1) and the Model developed by Pinto et al. [18] (DSS 2) to forecast coffee rust for the five municipalities in the Minas Gerais state, Brazil: (**a**) and (**b**) DSS 1 and DSS 2 applied in Carmo do Rio Claro, respectively; (**c**) and (**d**) DSS 1 and DSS 2 applied in Monte Santo de Minas, respectively; (**e**) and (**f**) DSS 1 and DSS 2 applied in Nova Resende, respectively; (**g**) and (**h**) DSS 1 and DSS 2 applied in Rio Paranaíba, respectively; (**i**) and (**j**) DSS 1 and DSS 2 applied in Serra do Salitre, respectively.

In the second phase of the modeling, we raised the hypothesis to elaborate meteorological variables, as described in phase 2 of the methodology, from temperature and precipitation data obtained from the five automatic meteorological stations installed by Cooxupé. Thus, 35 models were calculated, including the coffee harvest period and 35 models excluding it. The ten best-fitted models were selected with harvest period and nine were selected with no harvest period.

Based on the ten best-fitted models with the harvest period (Figure 4) and nine without this period (Figure 5), we verified the error, the residual or deviations concerning rust incidence observed in the field, which is the real data.

The models presented (Figures 4 and 5) were implemented in the interface to forecast rust incidence in the field and are being validated.

### 3.3. Phase 3: Expansion of the Warning System

In phase 2 of this study, the models using two regression equations (Table 5) as described in Section 3.2 were adjusted, and we selected those with the best adjustments for the five counties. These two models were selected based on the parameters of the regression equation, determination coefficient ($R^2$), Akaike information criterion (AIC), standard deviations and sum of the squared errors (Table 6).

**Table 6.** Qualitative index used to select the best models and describe the accuracy of the models for the Carmo do Rio Claro and Nova Resende municipalities, Minas Gerais state, Brazil.

| Model | Determination Coefficient [1] | AIC | Standard Deviation | Sum of Squared of Errors |
|---|---|---|---|---|
| Carmo do Rio Claro 15–30 DBRI | 0.67 *** | 4.11 | 8.95 | 155.71 |
| Nova Resende 15–30 DBRI | 0.56 *** | 0.76 | 1.24 | 1.71 |

[1] Regarding the significance of the test F: *** $p < 0.001$.

Afterward, we expanded those that presented promising field results to 15 more counties in the Minas Gerais state, Brazil. Currently, incidence sampling is being carried out in 50 areas, from which meteorological data are also being obtained to estimate rust values and thus issue phytosanitary warnings via the constructed interface. Notifications are issued from DSS 1 and 2 according to the methodology described above.

The incidence values calculated from the two adjusted models for Carmo do Rio Claro (Figure 6a) and Nova Resende (Figure 6b) obtained a good fit compared with the field data.

Since October 2020, these two models have been getting validated to forecast coffee rust incidence in ten counties apart from those where they were developed, totaling 15 municipalities (Figure 7), and from January 2021 in another 35 counties, totaling 50 places to send fungicide spray warnings.

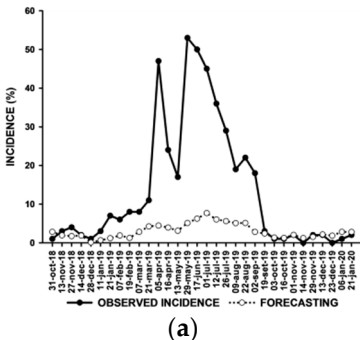
(a)

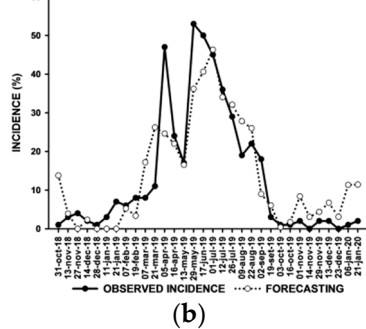
(b)

**Figure 4.** *Cont.*

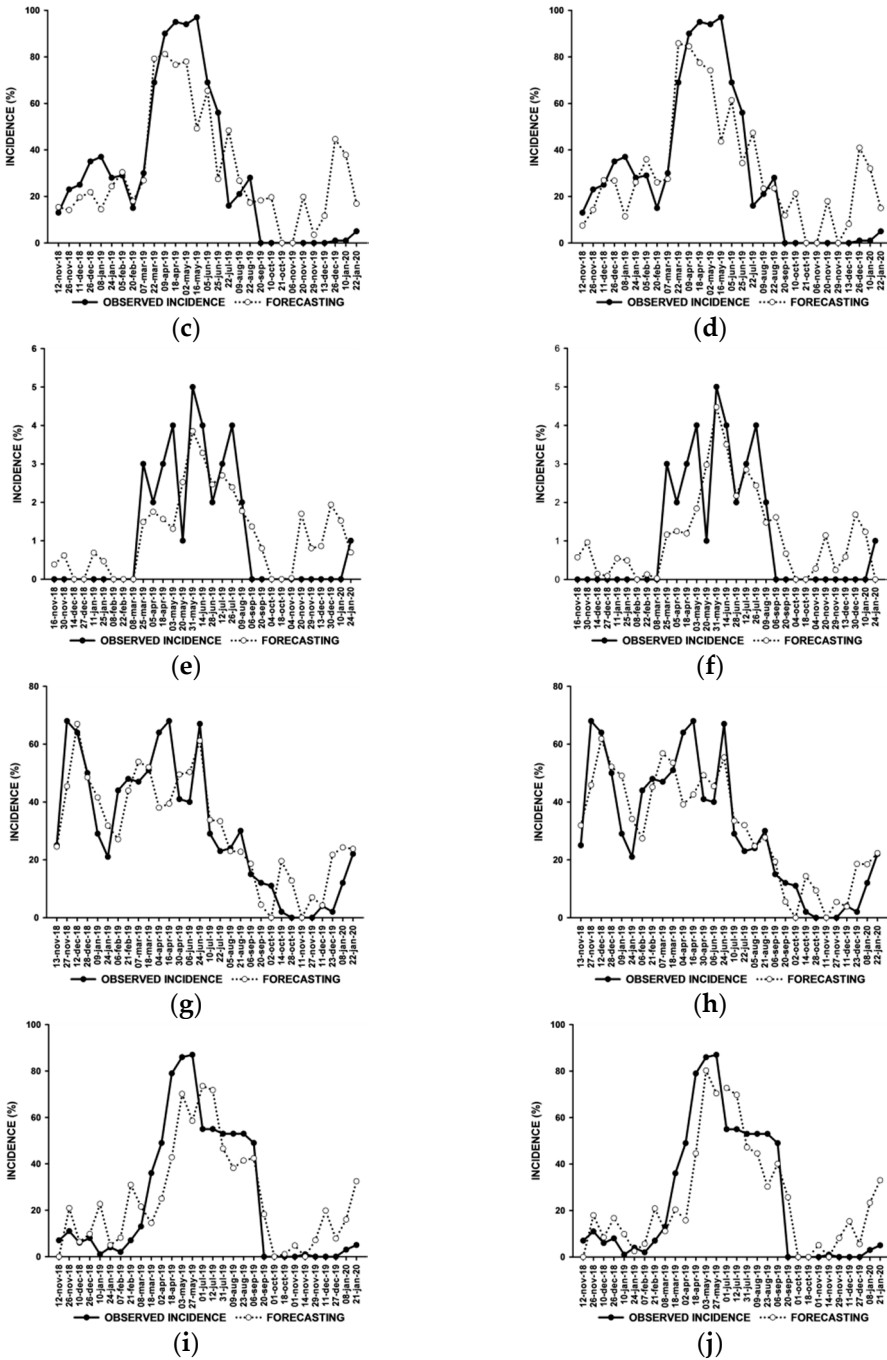

**Figure 4.** Performance of the best fitted models with forecast estimates of coffee leaf rust compared to field observed data: (**a**) Carmo do Rio Claro, model 15–45 DBRI (Days before rust incidence) with all variables; (**b**) Carmo do Rio Claro, model 2 15–45 DBRI excluding variables; (**c**) Monte Santo de Minas, model 15–45 DBRI with all variables; (**d**) Monte Santo de Minas, model 2 15–45 DBRI excluding variables; (**e**) Nova Resende, model 15–45 DBRI with all variables; (**f**) Nova Resende, model 2 15–45 DBRI excluding variables; (**g**) Rio Paranaíba, model 15–45 DBRI with all variables; (**h**) Rio Paranaíba, model 15–45 DBRI excluding variables; (**i**) Serra do Salitre, model 15–45 DBRI with all variables; (**j**) Serra do Salitre, model 2 15–45 DBRI excluding variables.

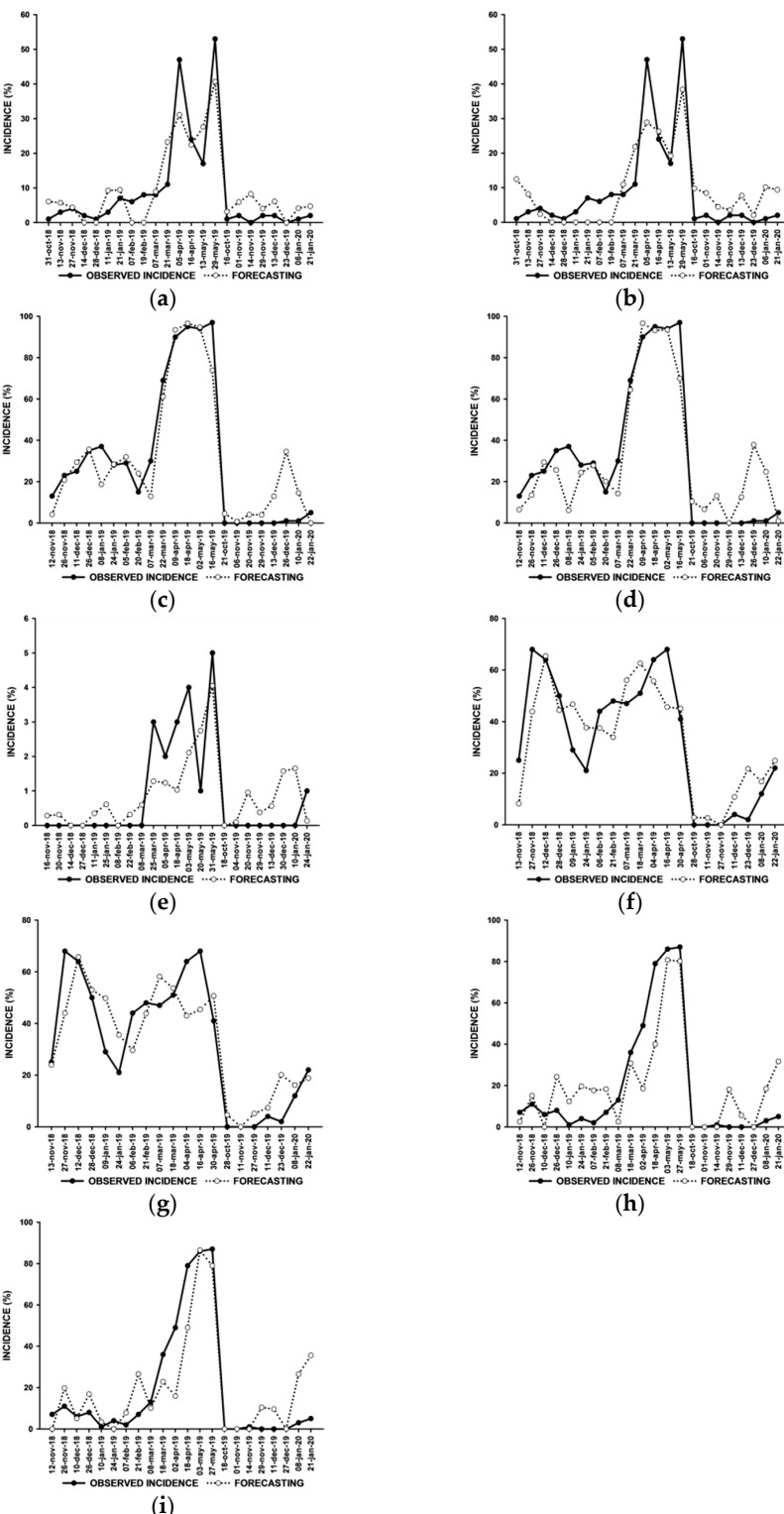

**Figure 5.** Performance of adjusted models excluding harvest for coffee rust forecast estimates compared to field observed data: (**a**) Carmo do Rio Claro, model 15–45 DBRI (Days before rust incidence) with all variables; (**b**) Carmo do Rio Claro, model 2 15–45 DBRI excluding variables; (**c**) Monte Santo de Minas, model 15–45 DBRI with all variables; (**d**) Monte Santo de Minas, model 2 15–45 DBRI excluding variables; (**e**) Nova Resende, model 2 15–45 DBRI with variable exclusion; (**f**) Rio Paranaíba, model 15–45 DBRI with all variables; (**g**) Rio Paranaíba, model 15–45 DBRI excluding variables; (**h**) Serra do Salitre, model 15–45 DBRI with all variables; (**i**) Serra do Salitre, model 2 15–45 DBRI excluding variables.

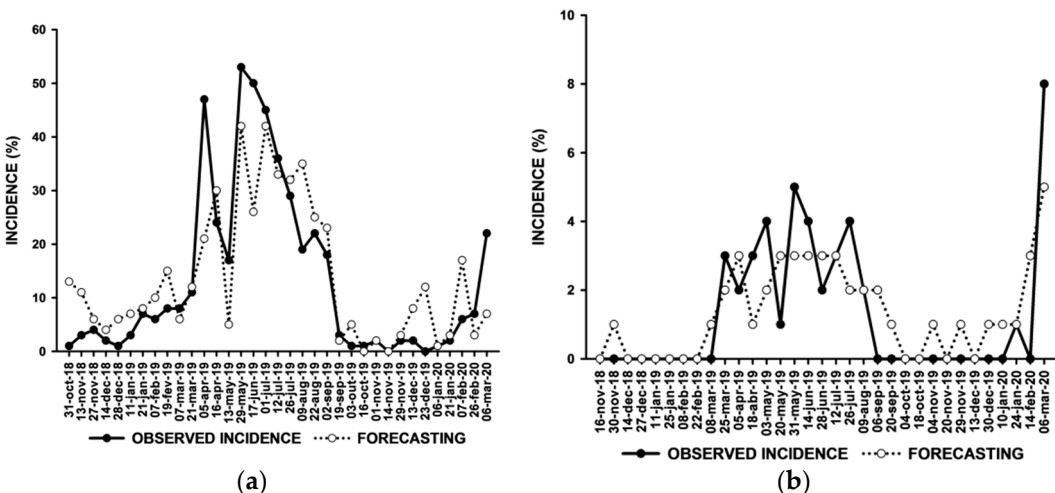

**Figure 6.** Coffee rust progress curve and prediction estimates from the models: (**a**) Carmo do Rio Claro model 15–30 DBRI (Days before rust incidence) excluding the Isolation time, Wind Speed, Dew point temperature and Wetness; (**b**) Nova Resende model 15–30 DBRI (Days before rust incidence) excluding Wind Speed and Dew point temperature.

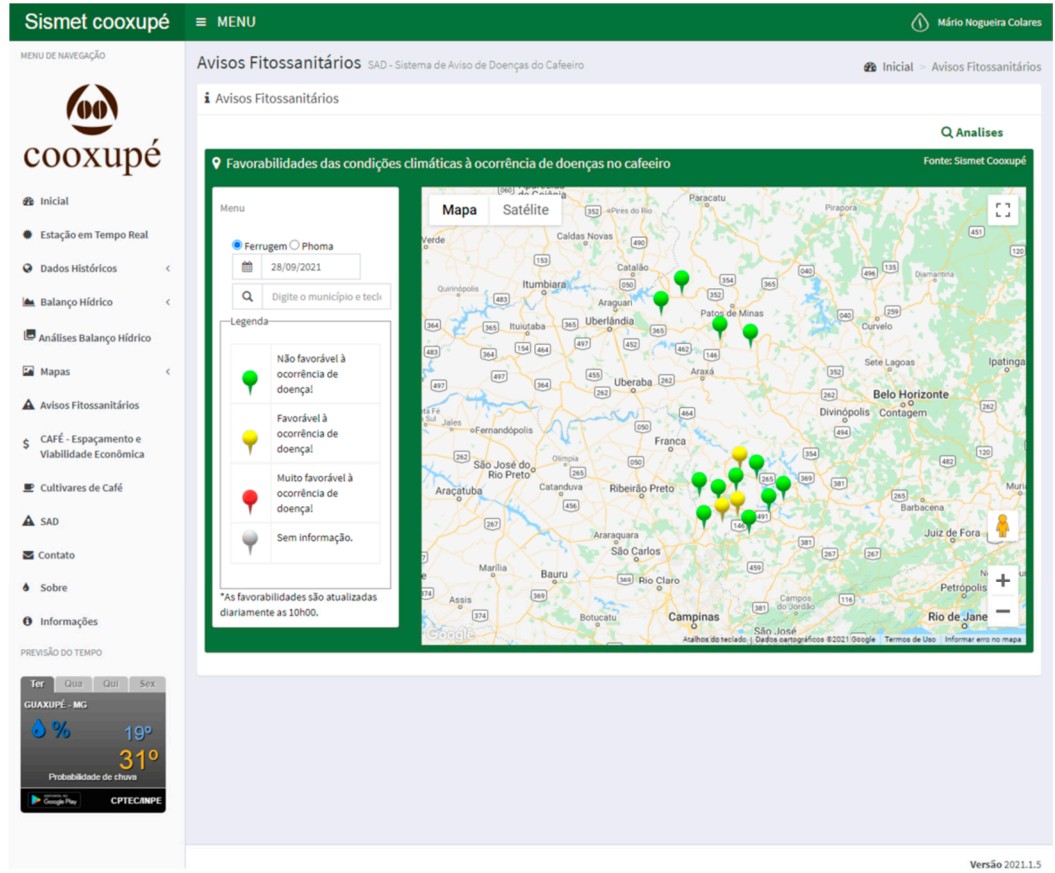

**Figure 7.** Real current phytosanitary forecast system interface of the Decision Support System (DSS) project showing the favorability of climatic conditions for the occurrence of coffee leaf rust according to the icon (green: not favorable to disease occurrence, yellow: favorable to disease occurrence, red: very favorable to disease occurrence and white: no information) in 15 municipalities in the states of Minas Gerais and São Paulo, Brazil. * Favorability are updated daily at 10 a.m.

Currently, the Disease Support System is predicting the incidence of coffee leaf rust in 15 municipalities. The expansion of the Disease Support System to other coffee producing municipalities in the states of Minas Gerais and São Paulo is expected in the future, and

thus the phytosanitary forecast interface can be visualized by coffee producers with the indication through the icon indicating the occurrence or not of coffee leaf rust.

## 4. Discussion

The models used for validation in the first phase did not obtain a good adjustment in all counties as expected. Several hypotheses emerged to improve the results for new model adjustments. Therefore, a second phase started using models adjusted with lagged variables of 15–30 DBRI since they were the models with the best results. There was an attempt to adjust at this stage and contemplate the premise of sending warnings of the disease forecast or warning 15 days beforehand. From the warning send, such time is needed for the producer to provide the spraying logistics based on the rust incubation period. That is, before pathogen sporulation or visualization of its signs. Nonetheless, it has already infected and colonized the coffee tree leaves [15,43].

In the second phase of adjustments, meteorological variables difficult to obtain in some meteorological stations were excluded, so that the formulas could use data collected from simpler systems or less sophisticated stations. It is even expected in the future to be collected on mobile devices such as smartphones. Thus, in the second phase of the model adjustments, new equations were obtained, two of which are under evaluation and serving to make predictions 15 days before the visualization of the disease signs in all counties where the experimental crops are located.

The selected models recorded coefficients of determination above 0.50 with lower values of errors, average errors, standard deviation and mean squared deviations; such coefficients of variation defined these disease forecasting models [33,74].

Yet in the second phase of the modeling, the meteorological variables temperature and precipitation were used to elaborate the averages and sums of temperatures and leaf moisture from 6 p.m. to 9 a.m. of the following day, with 6 h of free water or leaf wetness period as the minimum time required for infection [15].

In this phase of adjustments for some variables, one day was considered the interval from 12 h of one day to 11:59 h of the next day, as periods of leaf wetness occur between one day and another, especially at night, with no light and milder temperatures, ideal for germination and infection of *H. vastatrix*. Thus, due to its importance for urediniospore germination and pathogen penetration, the temperatures and leaf wetness period were collected and calculated daily [15]. These models were implemented for validation.

The meteorological and biological variables, by regression analysis, were considered to explain coffee leaf rust epidemics [75]. In this work, the most significant meteorological variables were identified using the Stepwise selection technique. Therefore, the model adjusted for Carmo do Rio Claro used the maximum and minimum temperature, precipitation, and minimum humidity as meteorological variables, whereas the model adjusted for Nova Resende used the average and minimum temperature, time of insolation and relative humidity. In each location, the weather condition is different, that is, there is variation in the environment. With this condition, there is variation in the intensity of the disease. Therefore, to explain the disease, for the model adjusted for Carmo do Rio Claro, the meteorological variables are different compared to the model adjusted for Nova Resende.

In both models, they were the meteorological variables with the best correlation with the incidence over time to explain and predict coffee rust 15 days in advance. Thus, these models were used to start the third phase or the expansion stage of the forecast system to 15 more counties where the climatological stations were located.

In this work, the period considered to obtain the meteorological variables to model the forecast of the coffee rust incidence was 15 to 30 days before the disease assessments or the visualization of the disease signs. This period was considered because it is related to the incubation period of coffee rust, ranging from 25 to 30 days [43,73] when the variables related to leaf moisture and temperature are crucial for the infection [75].

According to Campbell and Madden [33], phytosanitary forecasting or warning systems must be built with biological and environmental data to ensure criteria of reliability,

precision and accuracy. Based on meteorological and biological variables, the models adjusted in this work can estimate the coffee rust incidence in the field 15 days in advance. This period will allow better management to make decisions about the planning and logistics for disease control, thus issuing warnings when the coffee crop reaches 5% of rust incidence [43].

Another criterion highlighted by Campbell and Madden [33] is the usefulness of forecasting systems for crops where the diseases can be monitored and have effective control measures, which are met by coffee rust disease. Even more, when this extreme weather scenario is considered [3], with changes in moments of greatest incidence, especially at the beginning and end of the disease epidemic. The two best fitted models are easy to understand and interpret. This is an important aspect for their adoption or use by producers [76–78].

Phytosanitary forecasting or warning systems to aid decision-making for the control of coffee diseases were developed mainly for rust [18,43,44,51,55,56]. In works involving only regression modeling for this pathosystem, the best MLRM adjusted by Pinto et al. [18] and Hinnah et al. [44] obtained a $R^2$ of 80 and 86%, respectively, for the best models. In this work, the two models under validation developed for Carmo do Rio Claro and Nova Resende obtained a $R^2$ of 67 and 56% in these locations, respectively. However, both Pinto et al. [18] and Hinnah et al. [44] validated the selected models on the same data platform used to adjust the equations. However, in this work, the variables obtained from the meteorological stations over time are being used to forecast future disease intensity 15 days in advance, which is different from the dataset used in the elaboration of the models. In other words, the probability of error is greater since there is no vice or repetition of values already used in the calculation of equations.

The models adjusted for the counties of Carmo do Rio Claro and Nova Resende can be used for other regions, as they are based only on temperature and humidity. These meteorological variables have good spatial correlation and can be estimated, therefore, for other regions or areas of the same county using data from the meteorological stations [79]. This same model has precipitation as one of its parameters, this meteorological variable is limited to the place where it was obtained since it presents a significant spatial variability [79]. Thus, it is necessary to have rain gauges or stations distributed in the intended area so that the model's accuracy can be increased to emit the disease control warnings, in this case, of thousands of square kilometers, referring to the south and north of the state of Minas Gerais, the largest Arabica coffee producing region in the world.

These models, at present, are being validated in regions where they were adjusted, besides other locations where the meteorological variables can be obtained from already implemented stations. According to Campbell and Madden [33], it is essential to validate the warning system in other regions to make it credible. The results of forecasting models based on meteorological variables can be used in different regions from where the model was developed, but they must be tested over time. Validating these models, especially where the rust inoculum is abundant and the variability of both coffee cultivation methods and the pathogen biology is a reality, can adjust the parameter coefficients of regression equations to the meteorological conditions of those regions and make the model reliable.

## 5. Conclusions

In Brazil, forecasting systems to support the decision-making to control coffee rust are practically non-existent. This is probably because most of the models developed do not meet the criterion of simplicity [33], that is, anyone with a minimum knowledge should be able to interpret the results provided. The forecast system for rust based on meteorological variables applied in this study is simple.

Such variables are obtained at any meteorological station and can be calculated up to 15 days in advance, enough time for planning and managing fungicide spraying in small and large properties These were proven to be efficient in hitting the ideal time for spraying,

mitigating application risks in a less favorable infection time by the pathogen and thus optimizing control and increasing the sustainability of coffee growing.

The adjusted models for Carmo do Rio Claro and Nova Resende, Minas Gerais, Brazil, were the best at predicting the disease 15 days in advance.

The meteorological variables maximum, average and minimum temperature, hours of sunlight, precipitation, average and minimum relative air humidity were the main variables used to model the forecast of coffee rust.

The coffee rust forecast models showed great promise for issuing coffee plantation spraying warnings to the main producing regions of the state of Minas Gerais. However, the performance of these models for other coffee regions, with different climatic situations from those where they were adjusted, is being evaluated.

## 6. Final Considerations

Nowadays, the global concern regarding the presence of agrochemical residues in food has been reflected in the consumers' preferences for the trading and purchase of agricultural products. Allied to the growing increase in certifiers for the coffee production chain due to the concern and demand of the global population with the environment. Therefore, researchers share a common concern to use forecasting systems or phytosanitary alerts to optimize and even reduce fungicide applications.

The adjusted models for coffee rust forecast obtained as a function of meteorological variables can be used to issue phytosanitary warnings and minimize the economic, social and environmental impacts arising from the coffee rust incidence.

All models adjusted in this work that displayed good performance in the field will be implemented in the integrated meteorology and geographic information system (GIS) at the Cooxupé interface, with an expansion process through geoprocessing tools for the main coffee producing regions in Minas Gerais, Brazil. With this, all producers will have access to information from the phytosanitary alerts issued in the forecast system developed.

In the future, the results obtained in this work may serve as a basis for further coffee-growing research focused on economically and environmentally sustainable production.

**Author Contributions:** E.A.P.: Conception, elaboration of hypotheses, experiment methodology, data analysis, writing and review of the article; M.R.N.C.: Data analysis, development of regression models, data interpretation, drafting, writing and editing the article; É.R.d.S.: Survey and interpretation of field data; N.A.G.: Development of interface and software for integrating data collection with equations and issuing phytosanitary alerts, tabulation and data analysis; X.M.d.S.V.: Survey of field data and design of the experiment; M.d.C.A.: Elaboration of hypotheses, data analysis and integration of the modeling platform with the geographic information system. All authors have read and agreed to the published version of the manuscript.

**Funding:** This research was funded, M.R.N.C., support from the Coordination for the Improvement of Higher Education Personnel—CAPES (Ref. 88882.446551/2019-01). Grants for Edson Ampélio Pozza, support from the National Council for Scientific and Technological Development—CNPq (Ref. 310386/2017-9), the National Institute for Coffee Science and Technology—INCT Café and the Research Support Foundation of the State of Minas Gerais—FAPEMIG (Ref. CPQ APQ-03605-17), Co-operative of Coffee Growers in Guaxupé—Cooxupé and her partner Iharabras Chemical Industries.

**Institutional Review Board Statement:** Not applicable.

**Informed Consent Statement:** Not applicable.

**Data Availability Statement:** Not applicable.

**Acknowledgments:** The present work was the result of the joint efforts of the Laboratory of Epidemiology and Management of Plant Diseases in Phytopathology Department at the Federal University of Lavras (UFLA), the Cooperative of Coffee Growers in Guaxupé (Cooxupé) and Ihara Corporation, which contributed with Cooxupé from 2018 to 2021 and made the development of the research project possible. We thank all these partners, students and field technicians for carefully collecting the data and continuing with this arduous task, come rain or shine. Thank you very much to everyone.

**Conflicts of Interest:** The authors declare no conflict of interest. The funders had no role in the design of the study; in the collection, analyses, or interpretation of data; in the writing of the manuscript, or in the decision to publish the results.

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
