# Peer review of "Coffee Rust Forecast Systems: Development of a Warning Platform in a Minas Gerais State, Brazil"

_agronomy, doi:10.3390/agronomy11112284_

Round 1
Reviewer 1 Report
Dear Authors,
you can find my comments in the file attached.
All best

Author Response
Paper summary
What are the authors trying to achieve?
Authors are presenting the warning and forecast platform for coffee rust incidence fifteen days in advance. The work is carried out experimentally and numerically. First all environmental variables important for the plant disease development are collected, together with disease observations. Later existing models are validated on the observations.
Is this novel and within the remit of Agronomy?
Although systems like this have been developed, this one is new since it uses in situ observations of both meteorology and pathogen from different locations. It is in the scope of Agronomy.
Dear Authors,
I tend to review in quiet a conversational style, so apologies that there are many words here. The paper is unfortunately difficult to read. Sentences are too long and sometimes they lose their meaning. There are a lot of information given just in the Introduction. The part of the title “Past, Present and Future”, reminiscent of Review paper, not Article. And there are typos which are changing the meaning of the sentence. I understand that is the writing style, but for example your paper de Resende et al., 2021 (10.3390/agronomy11091865) is easier to read. All this is masking the work that has been done.
My comments fall under major revisions.
You can:
- consider shortening the sentences in order to improve the text flow and focus on the important information;
- change the title, since you are describing real work, I think that the “Past, Present and Future” can be removed, and instead you can be more specific;
Response: This suggestion was accepted and modified.
- In whole text you use word “validation” of the model. Here you need to be careful. In the methodology you speak about model adjustment – this is called model calibration, and it represent getting the right model parameters based on the observations. After the model calibration, model needs to be validated. The validation represent a comparison between the calculated values, and the observations. Data used for the model calibration cannot be used in the model validation. This is why in the beginning datasets are separated, to the data used for the training, and data used for the validation. Please have in mind this during your work and writing. This can be a problem if you have short sampling period, since then you have to train the model on the shorter dataset. One of the example of this is paragraph “The best models were validated from January to October 2020 in five counties. The lm function was used to adjust the MLRM, and the ggplot2 package was used to make graphs and the maps were implemented in the open access software “R” version 3.6.2 [72].” Here it looks like the validation was performed before calibration.
Response: You see, it is a concept. We can use model adjustment or calibration, but the word calibration should be used only in the model adjusted phase. This suggestion was accepted and modified in the paper.
“The lm function was used to adjust the MLRM, and the ggplot2 package was used to make graphs and the maps were implemented in the open access software “R” version 3.6.2 [72]. After the adjusted models, the best models were validated from February to October 2020 in five counties“.
- the graph that illustrate the data flow and creation of the models, elimination and criteria’s would be very informative and it will allow the reader to summarise the whole system.
Response: Dear reviewer, we appreciate the suggestion, the graph addition would be very useful for the paper indeed. However, due to the reduced response period stipulated by the Agronomy, unfortunately, we did not have enough time to fulfil the suggestion. In addition, we believe that even without this graph it will not change the paper good quality.
- Introduction
- with changing title consider removing the part of the text in the Introduction which only purpose is to explain the Past, Present and Future of the warning systems;
- if you decide to leave this part, make another title and put it under. With this the introduction will mirror the work which you are presenting.
- In the introduction be more oriented to the subject.
Response: This suggestion was accepted and modified. Title: “Coffee Rust Forecast Systems: Development of an Warning Platform in a Minas Gerais State, Brazil”.
- Methodology to construct forecast models and interface
Figure 1 – very difficult to read
Response: The figure is a Google Earth printout with the location of the 35 properties used for model validation.
2.1.2. Forecasting systems and spraying methodology
line 355 – and paragraph above. It is very difficult to understand which is DSS1, and which is DSS2. And there is then only one equation for DSS1.
Response: Correction accepted. The DSS 1 was based on a control level of 5%, and was validated when the fortnightly assessment of coffee rust incidence reached 5% in the field. The DSS 1 has no equation, is a methodology for disease control, and requires human resources for assessment. When it reached 5%, spraying was done. DSS 2 based on the equation proposed by Pinto et al., (2002). DSS 2 also used the 5% control level, but used the equation to calculate the 5% incidence level.
Table 3 – Please introduce symbols for the values. There is established symbology for the mereorological values that you use, like average temperature can be Tavg. With this formula in Table 4 will look nicer.
Response: This suggestion was accepted and modified.
We know that environment and appropriate plant developing stage are controlling the disease development.
2.3. Phase 3: Expansion of the warning system with new models
“Beyond the two models selected in phase 2 for DSS 1, DSS 2 - a Meteorological Model is also being used to forecast or send the disease warning 15 days in advance in 15 counties.” what is Meteorological Model
Response: Meteorological Model is the DSS 2. The paragraph was rewritten.
“Beyond the two models selected in phase 2 for to substitute DSS 1, the DSS 2 denomined Meteorological Model is also being used to forecast or send the disease warning 15 days in advance in 15 counties.”
- Results
3.1. Phase 1: Validation of the models proposed by Pinto et al. [18]
The results from the Figure 3 should be explained. It is not enough to say that there was a difference. Also the difference should be quantified.
Response: The DSS 1 was replaced, because it did not performed good results in controlling the disease. A paragraph was inserted in the paper explaining its replacement.
“The DSS 1 was replaced. Because in the field validation period, the 5% control level methodology presented failures in the control of the disease, whaen compared to the other models being validated. And also this methodology needs human resources available for the assessments every fifteen days.”
Figures 3-6 should be make more readable. Text is too small.
Response: By increasing the size of the graphics, figure 3-5 will be on two pages. The size of figure 6 was increased.
3.2. Phase 2: Adjustment of forecasting models in five different counties in the state of Minas Gerais, Brazil
- Discussion
“Therefore, the model adjusted for Carmo do Rio Claro used the maximum and minimum temperature, precipitation, and minimum humidity as meteorological variables, whereas the model adjusted for Nova Resende used the average and minimum temperature, time of insolation and relative humidity.“ - This should be explained.
Response: This suggestion was accepted. It was explained in the paper.
“In each location, the weather condition is different, that is, variation in the environment. With this condition, there is variation in the intensity of the disease. So, therefore, to explain the disease, for the model adjusted for Carmo do Rio Claro the weather variables are different compared to the model adjusted for Nova Resende.“
There are a lot of important findings mention for the first time in this part of the paper. For me this part was the most interesanting, since you have explained parts that I had problems with, like use of the statistical model on other locations.
Try to make it shorter, more concrete and to avoid saying the same thing twice.
Finally I want to reiterate that there is a lot of interesting work in this paper. So thinking through exactly what is novel, why that is important. Very happy to take a future look at this.
Reviewer 2 Report
This study aims to develop a warning platform for the coffee rust incidence, 15 days in advance, to validate and regionalize Multiple Linear Regression Models from regionalized meteorological variables for the main coffee producing regions and biomes in the state of Minas Gerais, Brazil.
A lot of research on the previous works has been done over the past decades and they were reported in the introduction. The author should present the originality of this study, which is different from previous studies on the forecast systems. The materials and method, and results appears neglected and lacking in detail useful for understanding the work done. Only the final results are presented, skipping the intermediate steps useful for understanding the method used.
Authors may pay attention to the following main suggestions.
Title
Please consider changing the title from “Coffee Rust Forecast Systems: Past, Present and Future”, to e.g., “Coffee Rust Forecast Systems: Development of an Alert Platform in the State of Minas Gerais, Brazil”
Introduction
Consider reducing the length of the section or dividing it into subsections.
Line 87-99-etc.: please change “warning … system” with “alert…system” to be consistent with the abstract.
Line 218: please, delete “model” after “…the model of Pinto et al. [18]”.
M&M
In general, the section is very confusing, and although it was divided into 3 phases it is difficult to understand the succession of events and operations. The experimental test plan, with the 5 treatments, was explained in a superficial and incomplete way. The equations underlying the forecasting methods need to be improved in presentation. Often the equations / methods were presented without explanation.
Lines 303-305: introduce the paragraph appropriately, it is not clear.
Line 311: better define what “35 points” are: counties, properties, points, farms?
Section 2.1.2.
The 2 models are unclear, DSS 1 comes with equation, DSS 2 no, why?
Eq(1) and (2): eliminate the comma at the end of the equations.
Line 449: it is not clear what “item a” is, did you mean “item 1”?
Section 2.2.
Phase 2: many methods and systems are introduced to adjust the model but there is no confirmation in the results.
Line 416: please consider writing the entire MLRM. Written only on the abstract.
Phase 3. line 492: what are the three equations mentioned?
Line 512: please consider changing “southem” with “southern”
Results
The step in finding the best model (s) is not clear. The feeling is that there was an intention to create many models by selecting the one with the lowest error. However, the evaluation of errors is not reported. The problem is that the correct model for that given place, with those climatic / meteorological conditions, at that time may not be replicable and cannot be used as a general model. In my opinion, a forecast model that needs to be corrected every time a variable changes is not a consistent model. Each location has a custom model. So, every time you change locations, do you have to revalidate the model? How many correct models there are, it is not clear!
Section 3.2: It seems that the 4 models correspond to those written in line 420. But then the models become 35 out of 5 counties. Would that be 4 + 2 (old) x 5 = 35?
I recommend rewriting the paragraph so that it can be better understood.
Lines 609 and 612: please avoid writing "we".
Caption Figure 6: missing space between “incidence)” and “excluding”.
Caption Figure 7: the word "Real" is in bold. Consider consistency with the style of the text.
Discussion
Section too long. Too many quotes, it must be a discussion of the work. For instance, lines 707-726 in my opinion are unnecessary.
From line 744 onwards some data relating to the results are introduced. Consider revising the section by moving the parts corresponding to the results to the corresponding section.
Conclusion
Lines 781-790 can be moved to this section; it feels more like a conclusion rather than a discussion.
Author Response
This study aims to develop a warning platform for the coffee rust incidence, 15 days in advance, to validate and regionalize Multiple Linear Regression Models from regionalized meteorological variables for the main coffee producing regions and biomes in the state of Minas Gerais, Brazil.
A lot of research on the previous works has been done over the past decades and they were reported in the introduction. The author should present the originality of this study, which is different from previous studies on the forecast systems. The materials and method, and results appears neglected and lacking in detail useful for understanding the work done. Only the final results are presented, skipping the intermediate steps useful for understanding the method used.
Authors may pay attention to the following main suggestions.
Title
Please consider changing the title from “Coffee Rust Forecast Systems: Past, Present and Future”, to e.g., “Coffee Rust Forecast Systems: Development of an Alert Platform in the State of Minas Gerais, Brazil”
Response: This suggestion was accepted and modified. Title: “Coffee Rust Forecast Systems: Development of an Warning Platform in a Minas Gerais State, Brazil”.
Introduction
Consider reducing the length of the section or dividing it into subsections.
Response: This suggestion was accepted and modified. The indroduction was reduced.
Line 87-99-etc.: please change “warning … system” with “alert…system” to be consistent with the abstract.
Response: Dear reviewer, the term used in paper is “warning”. So we will leave “warning” and replace “Alert” in the abstract.
Line 218: please, delete “model” after “…the model of Pinto et al. [18]”.
Response: This suggestion was accepted.
M&M
In general, the section is very confusing, and although it was divided into 3 phases it is difficult to understand the succession of events and operations. The experimental test plan, with the 5 treatments, was explained in a superficial and incomplete way. The equations underlying the forecasting methods need to be improved in presentation. Often the equations / methods were presented without explanation.
Lines 303-305: introduce the paragraph appropriately, it is not clear.
Response: The use of predictive models is an alternative to optimize and rationalize the use of fungicides. Therefore, were evaluated the viability of two prediction models developed by Pinto et al. [18]. Since the forecast models were developed with meteorological data from Lavras county, Minas Gerais state, Brazil, other geographic locations were incorporated to validate the models. Then, the study was divided into three phases.
Line 311: better define what “35 points” are: counties, properties, points, farms?
Response: The 35 points are 35 properties. This suggestion was accepted to modify.
Section 2.1.2.
The 2 models are unclear, DSS 1 comes with equation, DSS 2 no, why?
Response: This suggestion was accepted to modified.
Eq(1) and (2): eliminate the comma at the end of the equations.
Response: This suggestion was accepted.
Line 449: it is not clear what “item a” is, did you mean “item 1”?
Response: Yes, is “item 1”. This suggestion was accepted to modified.
Section 2.2.
Phase 2: many methods and systems are introdução to adjust the model but there is no confirmation in the results.
Line 416: please consider writing the entire MLRM. Written only on the abstract.
Response: This suggestion was accepted to modified.
Phase 3. line 492: what are the three equations mentioned?
Response: This suggestion was accepted to modified. That is, the equations were mentioned in the text.
The three equations mentioned are:
The 2 best fit equations in phase 2:
- Model Carmo do Rio Claro: Y = -304.78667634*** + 13.16506156 Tmax*** - 13.91295446 Tmin*** - 0.07066871 P + 3.42680986 RHmin*** and;
- Model Nova Resende: Y = 3098934*** - 1.3387465 Tavg** + 1.2618530 Tmin** - 1.0714839 IH** - 0.2205387 WH**
The third equations is proposed by Hinnah et al., 2020:
- Model Hinnah: y = (-1.293+(0.019*Tmin(30-60d) + (0.017*RH(30-60d)
Line 512: please consider changing “southem” with “southern”
Response: It was a typing error. This suggestion was accepted and modified.
Results
The step in finding the best model (s) is not clear. The feeling is that there was an intention to create many models by selecting the one with the lowest error. However, the evaluation of errors is not reported. The problem is that the correct model for that given place, with those climatic / meteorological conditions, at that time may not be replicable and cannot be used as a general model. In my opinion, a forecast model that needs to be corrected every time a variable changes is not a consistent model. Each location has a custom model. So, every time you change locations, do you have to revalidate the model? How many correct models there are, it is not clear!
Response: Dear reviewer, it is not necessary to revalidate the model when changing locations, because the model is being validated for locations other than the one where the model was fitted. There are two models that have been adjusted and are being validated in the field.
Section 3.2: It seems that the 4 models correspond to those written in line 420. But then the models become 35 out of 5 counties. Would that be 4 + 2 (old) x 5 = 35?
I recommend rewriting the paragraph so that it can be better understood.
Response: Dear reviewer, the four models correspond to those written in item 1. However, these models were fitted with meteorological variables that are difficult to obtain from less sophisticated meteorological stations. These models have been discarded. In item 2, it is mentioned to fit models by 15-30 days before the disease assessment and therefore 35 models are obtained and of which 10 best models were selected and finally 2 most promising models to prevent rust incidence.
Lines 609 and 612: please avoid writing "we".
Response: This suggestion was accepted and modified.
Caption Figure 6: missing space between “incidence)” and “excluding”.
Response: This suggestion was accepted and modified.
Caption Figure 7: the word "Real" is in bold. Consider consistency with the style of the text.
Response: This suggestion was accepted and modified.
Discussion
Section too long. Too many quotes, it must be a discussion of the work. For instance, lines 707-726 in my opinion are unnecessary.
Response: This suggestion was accepted and modified.
From line 744 onwards some data relating to the results are introduced. Consider revising the section by moving the parts corresponding to the results to the corresponding section.
Response: This suggestion was accepted and modified.
The two best fitted models are easy to understand and interpret. This is an important aspects for their adoption or use by producers.
Conclusion
Lines 781-790 can be moved to this section; it feels more like a conclusion rather than a discussion.
Response: This suggestion was accepted and added to this section.
Reviewer 3 Report
Review of “Coffee Rust Forecast Systems: Past, Present and Future”.
The authors aimed to develop an alert platform for coffee rust incidence fifteen days in advance. The topic of this paper is interesting and useful, and the forecast system will be valuable for Coffee Rust control. However, I still have some concerns before the paper can be accepted for publication.
- The paper aimed to develop an alert platform for coffee rust incidence, however the title of this paper looks like a REVIEW article. I suggest the authors modify it, to avoid misunderstanding for the first look.
- The authors used multiple linear regression models to establish the Forecast System, and the results showcase that the forecast accuracy can probably be improved. Have the authors considered machine learning methods? The authors may introduce machine learning (especially deep learning) in future studies.
- How do the authors obtain the meteorological data? Also, how do the authors determine the lagging effect?
- Section 3.3: Figure 6 provides the Coffee rust progress curve and prediction estimates from the models. I still suggest the authors give some quantitative index to describe the model accuracy for easier understanding.
- Introduction: The authors present a detailed review and analysis of coffee rust forecast systems. Can the authors divide this part into several sub-parts? Thus, it is more layered and better organized, and clearer for readers to understand.
Author Response
Review of “Coffee Rust Forecast Systems: Past, Present and Future”.
The authors aimed to develop an alert platform for coffee rust incidence fifteen days in advance. The topic of this paper is interesting and useful, and the forecast system will be valuable for Coffee Rust control. However, I still have some concerns before the paper can be accepted for publication.
- The paper aimed to develop an alert platform for coffee rust incidence, however the title of this paper looks like a REVIEW article. I suggest the authors modify it, to avoid misunderstanding for the first look.
Response 1: This suggestion was accepted and modified.
Title: “Coffee Rust Forecast Systems: Development of an Warning Platform in a Minas Gerais State, Brazil”.
- The authors used multiple linear regression models to establish the Forecast System, and the results showcase that the forecast accuracy can probably be improved. Have the authors considered machine learning methods? The authors may introduce machine learning (especially deep learning) in future studies.
Response 2: No machine learning methods were considered. Because at this first moment it was considered to obtain simple regression models. However, in future studies it is possible to deep learning methods.
- How do the authors obtain the meteorological data? Also, how do the authors determine the lagging effect?
Response 3: Environmental data are obtain from the automatic stations model Davis Vantage Pro® in the counties and transferred to the Weather Link® software located in the Cooperative of Coffee Growers in Guaxupé (Cooxupé). The lagging effect was performed by average/accumulated values of weather variables were counted in days lagged from the date of disease occurrence in the field of 2-4, 4-7, 7-15, 15-30 and 15-45 days before rust incidence. For example, the period of weather data from 15 to 30 days before to disease occurrence is considered for disease warning, and there is a gap of 15 days which is not considered for the calculation.
- Section 3.3: Figure 6 provides the Coffee rust progress curve and prediction estimates from the models. I still suggest the authors give some quantitative index to describe the model accuracy for easier understanding.
Response 4: This suggestion was accepted and added to the paper in a table.
“These two models were selected based on the regression equation parameters, determination coefficient values, akaike information criterion values, standard deviations and sum of the squared errors.”
- Introduction: The authors present a detailed review and analysis of coffee rust forecast systems. Can the authors divide this part into several sub-parts? Thus, it is more layered and better organized, and clearer for readers to understand.
Response 5: I agree with the reviewer in reducing the introduction for better understanding for the readers. This suggestion was accepted and modified.
Round 2
Reviewer 1 Report
Dear Authors,
thank you for replying to all my comments. I am still not satisfied with the figure quality, but if editor agree it can be like this. I understand that the figure is from Google, but there are also ways to make the map by using some other tools, or by improving the quality of existing one.
All best
Reviewer 2 Report
The suggestions were considered in the review.
The manuscript can be accepted in its present form.